# CRISPR/Cas9-Mediated Models of Retinitis Pigmentosa Reveal Differential Proliferative Response of Müller Cells between *Xenopus laevis* and *Xenopus tropicalis*

**DOI:** 10.3390/cells11050807

**Published:** 2022-02-25

**Authors:** Karine Parain, Sophie Lourdel, Alicia Donval, Albert Chesneau, Caroline Borday, Odile Bronchain, Morgane Locker, Muriel Perron

**Affiliations:** Paris-Saclay Institute of Neuroscience, CNRS, Université Paris-Saclay, CERTO-Retina France, 91400 Saclay, France; karine.parain@universite-paris-saclay.fr (K.P.); sophie.lourdel@universite-paris-saclay.fr (S.L.); alicia.donval@universite-paris-saclay.fr (A.D.); albert.chesneau@universite-paris-saclay.fr (A.C.); caroline.borday@universite-paris-saclay.fr (C.B.); odile.bronchain@universite-paris-saclay.fr (O.B.)

**Keywords:** CRISPR/Cas9, photoreceptors, retina, rhodopsin, Müller cells

## Abstract

Retinitis pigmentosa is an inherited retinal dystrophy that ultimately leads to blindness due to the progressive degeneration of rod photoreceptors and the subsequent non-cell autonomous death of cones. *Rhodopsin* is the most frequently mutated gene in this disease. We here developed *rhodopsin* gene editing-based models of retinitis pigmentosa in two *Xenopus* species, *Xenopus laevis* and *Xenopus tropicalis*, by using CRISPR/Cas9 technology. In both of them, loss of *rhodopsin* function results in massive rod cell degeneration characterized by progressive shortening of outer segments and occasional cell death. This is followed by cone morphology deterioration. Despite these apparently similar degenerative environments, we found that Müller glial cells behave differently in *Xenopus laevis* and *Xenopus tropicalis*. While a significant proportion of Müller cells re-enter into the cell cycle in *Xenopus laevis*, their proliferation remains extremely limited in *Xenopus tropicalis*. This work thus reveals divergent responses to retinal injury in closely related species. These models should help in the future to deepen our understanding of the mechanisms that have shaped regeneration during evolution, with tremendous differences across vertebrates.

## 1. Introduction

The degeneration of photoreceptors in retinal diseases, such as age-related macular degeneration or retinitis pigmentosa, is irreversible in humans and therefore leads to visual impairment. The virtual incapacity of mammals to replace lost retinal neurons dramatically contrasts with the extraordinary self-repair properties exhibited by fish or amphibians. Understanding spontaneous regenerative mechanisms in such species may help in designing therapeutic strategies to trigger similar processes in patients.

Müller cells are the major glial cell type of the retina that provide homeostatic, metabolic and structural support to retinal neurons [1]. These glial cells were identified as the cellular source of retinal regeneration in zebrafish [2]. Contrasting with their mammalian counterparts, whose regenerative properties are limited, zebrafish Müller glia behave as genuine multipotent stem cells, able to replace all retinal cell types upon injury. We recently discovered that this holds true in the frog *Xenopus laevis* (*X. laevis*) as well [3]. In this species, mechanical injury or conditional rod cell ablation triggers Müller cells to re-enter into the cell cycle and regenerate lost photoreceptors. Despite the absence of spontaneous regeneration in mammals, in vitro and in vivo studies demonstrated that mammalian Müller cells can also reprogram and produce new neurons in response to injury, provided treatment with appropriate growth factors or following overexpression of specific proneural genes [4,5,6,7]. Although much progress has been made these past few years in identifying genes and signalling pathways sustaining Müller cell regeneration [5,6], the molecular bases that account for interspecies differential potential are still largely unknown.

Beyond the tremendous variability of retinal repair efficiency among vertebrate species, diverse cellular sources can drive the regeneration process, whose mobilization can depend on the injury paradigm applied. In *Xenopus* in particular, the retinal pigmented epithelium (RPE) can reprogram following retinectomy and regenerate the neuroretina [8]. Besides, new retinal neurons can also originate from the ciliary marginal zone (CMZ), a small peripheral region of the adult eye that contains mitotically active retinal stem cells and primarily sustains continuous tissue growth along the animal life. Interspecies differences have here again been reported, with *X. laevis* and *Xenopus tropicalis* (*X. tropicalis*) responding differently following full removal of the retina. In *X. laevis*, the central and peripheral parts of the retina are regenerated from RPE and CMZ cells, respectively [8]. In contrast, only CMZ cells contribute to the whole regenerative process in *X. tropicalis* [9]. Comparing different species with various abilities and distinct modes of regeneration will undoubtedly help in identifying the molecular cues that either promote or constrain retinal self-repair. With such cross-species analyses in mind, and given the differential regenerative responses of *X. laevis* and *X. tropicalis*, we wondered whether *X. tropicalis* is also able to recruit Müller cells in case of targeted photoreceptor degeneration.

Undertaking comparative analyses of retinal regeneration in two distinct species requires establishing a common injury paradigm. Here, we used CRISPR-Cas9-mediated *rhodopsin* (*rho*) gene editing to generate models of retinitis pigmentosa in both *X. laevis* and *X. tropicalis*. The *rho* gene encodes the rod visual pigment, and its mutations in humans are the most common causes of autosomal-dominant retinitis pigmentosa, accounting for 20–30% of all cases [10]. This disease leads first to rod cell dysfunction and death, which secondarily compromises cone survival. We found that *rho* gene editing in both *Xenopus* species nicely models the main features of the disease, including severe rod degeneration and subsequent cone morphology deterioration. Besides, we show that although a subpopulation of Müller glial cells efficiently re-enter into the cell cycle in *X. laevis* following photoreceptor degeneration, the proliferative response of *X. tropicalis* Müller cells is extremely limited. This work thus further highlights how regeneration cellular process can tremendously vary in vertebrates, even among closely related species.

## 2. Materials and Methods

### 2.1. CRISPR sgRNA Design and RNA Synthesis

The sgRNA sequence targeting both *X.tropicalis* and *X.laevis rhodopsin* exon 1 was designed using the Crispor website (http://crispor.tefor.net/ accessed on 1 April 2015). This sgRNA was similar to that previously described [11], although with an additional nucleotide on both 5′ and 3′ end. The sequence complementary to the target DNA is indicated in Figure 1A. *Rho* sgRNA targeting sequence was cloned by PCR amplification into the pUC57-Simple-gRNA backbone (a gift from Yonglong Chen, Addgene plasmid #51306; http://n2t.net/addgene:51306 accessed on 1 January 2015; RRID:Addgene_51306) [12]. The primers used are the following:

Forward:

CATCCGATACCCAATTAATACGACTCACTATAGGCTCTGCTAAGTAATACTGA GTTTTAGAGCTAGAAATAGCAAGTTAAAATAAGGCTAG

Reverse:

CTAGCCTTATTTTAACTTGCTATTTCTAGCTCTAAAACTCAGTATTACTTAGCAGAGCCTATAGTGAGTCGTATTAATTGGGTATCGGATG

PCR product was digested with DpnI (20 U) for 1 h, purified and sent to sequencing. To synthetize the *rho* sgRNA, the construct was digested with DraI, transcribed using T7 MEGAShortscript kit (Invitrogen, Waltham, MA, USA) and purified with Nucleospin miRNA kit (Macherey-Nagel, Düren, Germany). To produce capped Cas9 mRNA, plasmid pCS2-3xFLAG-NLS-SpCas9-NLS (a gift from Yonglong Chen, Addgene plasmid #51307; http://n2t.net/addgene:51307 accessed on 1 January 2015; RRID:Addgene_51307) [12] was linearized with NotI, transcribed using mMessage mMachine SP6 kit (Invitrogen) and purified with the Nucleospin RNA clean-up XS kit (Macherey-Nagel).

### 2.2. Animals and Micro-Injections

*X. laevis* and *X. tropicalis* tadpoles were obtained by conventional procedures of in vitro or natural fertilization, staged according to Nieuwkoop and Faber [13] and raised at 18–20 °C. *Rho* sgRNA (250 to 500 pg) and *Cas9* mRNA (600–1200 pg) or Cas9 protein (2.5–5 ng for *X. laevis* and 1–2 ng for *X. tropicalis*) were injected at the one-cell stage. Cas9 protein was ordered from New England Biolabs (Ipswich, MA, USA, #M0646T). When delivered alone at the aforementioned doses, neither *Cas9* mRNA, Cas9 protein nor *rho* sgRNA showed toxicity, based on survival rates and morphology of injected embryos, compared to non-injected ones. For Cas9 protein, we recommend a maximum dose of 5 ng per *X. laevis* embryo and 2 ng per *X. tropicalis* one. In our experiments, these doses resulted in efficacies similar to that obtained with 600–1200 pg of *the* corresponding mRNA.

### 2.3. Genotyping of Embryos

Genomic DNA was extracted from a single embryo or from a mixture of 3 to 5 embryos. The targeted region was then amplified by PCR. Primers used are indicated in Appendix A. Amplicons were purified using PCR clean-up kit (Macherey-Nagel) and sequenced directly or after cloning in PCRII Vector (TA cloning Kit Dual Promotor, Life Technologies, Carlsbad, CA, USA) for definitive genotyping.

Chromatograms obtained from direct sequencing of PCR products were analysed with the TIDE (Tracking of Indels by DEcomposition) software [14] to identify insertions and deletions (indels).

### 2.4. BrdU Labelling and Tissue Preparation

Tadpoles were immersed in a solution containing 0.5 to 1 mM BrdU (5′-bromo-2′-deoxyuridine, Roche, Bâle, Switzerland) for 3 days before fixation. At the indicated time, they were anesthetized in 0.4% MS222 and fixed in 4% paraformaldehyde for 2–3 h at room temperature. Paraffin-embedded tadpoles were sectioned (11 µm) on a Microm HM 340E microtome (MM-France, Brignais, France).

### 2.5. Immunofluorescence, H&E Staining

Immunostaining was performed on paraffin sections using standard procedure. Antigen retrieval was performed by boiling the sections in 10 mM sodium citrate and 0.05% Tween20 for 9 min. For BrdU immunostaining, slides were also incubated in 2 N HCl for 45 min at room temperature, as a second unmasking method. After a 1-h blocking step in 10% goat serum (Life technologies), 5% BSA (Sigma, Saint-Louis, MO, USA), 0.2% Tween20 (Sigma), slides were incubated with primary antibodies overnight at 4 °C, rinsed 3 times in 1X PBS supplemented with 0.1% Triton X100 for 10 min, and then incubated for 2 h at room temperature with secondary antibodies. All used antibodies are listed in Appendix A. Cell nuclei were counterstained with Hoechst (Sigma), and coverslips were mounted using Fluorsave Reagent (Millipore, Burlington, MA, USA). Hematoxylin and eosin (H&E) staining was performed according to standard procedures.

### 2.6. Microscopy, Quantification, and Statistical Analysis

Fluorescence and brightfield images were obtained with an Imager M2 microscope (Zeiss). Images were processed using Zen (Zeiss, Oberkochen, Germany) and Photoshop CS5 (Adobe) software. Quantification of labelled cells was performed by manual counting. Three to ten sections per retina were analysed, and an average number was calculated. For measurements of staining areas, 1 to 3 sections were analysed per retina; the total labelled surface was quantified using Image J software after setting an optimal intensity threshold. For thickness quantification, the length of 10 labelled photoreceptors was measured on one representative section per retina. All experiments were performed at least in duplicate. Figures show the results from one representative experiment. Statistical analyses were performed using the nonparametric Mann–Whitney test to determine statistical differences between samples. Mann–Whitney result: ns for non-significant; * *p* < 0.05; ** *p* < 0.01; *** *p* < 0.001; **** *p* < 0.0001.

### 2.7. Ethics Statement

All animal care and experimentation were conducted in accordance with institutional guidelines, under the institutional license C 91-471-102. The study protocol was approved by the institutional animal care committee, with the reference number APAFIS#5938-20160704l3l 04812 v2.

## 3. Results

### 3.1. Generation of CRISPR/Cas9-Induced Rhodopsin Mutations in X. tropicalis

Efficient CRISPR/Cas9-mediated editing of the *rhodopsin* (*rho*) gene was previously reported in both *Xenopus laevis* [11] and zebrafish [15] and was shown to generate features of retinitis pigmentosa. In order to develop a similar model in *X. tropicalis*, a diploid *Xenopus* species displaying a single *rho* gene, we designed a specific single guide RNA (sgRNA) targeting exon 1 (Figure 1A). Then, 500 pg of *rho* sgRNA was injected into one cell stage *X. tropicalis* fertilized oocytes, together with 600 pg of *cas9* mRNA. The resulting embryos are hereafter named crispants. Control embryos were injected with *cas9* mRNA alone. In order to assess the efficiency of genome editing, we amplified the targeted region from two independent mixtures of five tadpoles. PCR amplicons were then cloned and sequenced. Among analysed sequences, 73% (8/11) exhibited insertion–deletion (indels). Four of them led to frameshifts (probably resulting in truncated proteins or nonsense-mediated decay), and four others generated amino acid deletions (Figure 1B). This percentage is similar to previous CRISPR/Cas9 reports in *X. tropicalis* [12,16,17] and reveals an efficient *rho* gene editing in this species using this technology.

### 3.2. Characterization of Photoreceptor Degeneration in X. tropicalis rho Crispants

We next aimed at characterizing the phenotype of *rho* crispant individuals. Retinal morphology was first analysed by Hematoxylin and Eosin (H&E) staining on stage 45 tadpoles. When injected at a dose of 250 pg, with either Cas9 mRNA or protein, the sgRNA did not trigger any major defects in the organization of the different retinal cell layers. However, tadpoles that received 500 pg displayed a reduction or even an absence of photoreceptor outer segments, indicative of cell degeneration (Figure 2A,B). We then examined Rhodopsin expression. Of note, considering the type of indels generated by Cas9 protein (Figure 1B) and the mosaicism of crispants [18], we inferred that at least part of mutant cells may still exhibit immunoreactivity. This analysis revealed a progressive loss of Rhodopsin expression along with development. In stage 40 crispant retinas (a stage that corresponds to the end of embryonic retinogenesis), the number of Rhodopsin-positive cells was decreased by 33% compared to controls (Figure 2C,D). Rhodopsin staining dramatically declined thereafter, with only rare labelled cells persisting in the central retina at stage 45 (Figure 2E,F). Such temporary Rhodopsin detectability suggests that mutant cells are either heterozygous and/or produce a mutated or truncated protein still recognizable by the antibody. Nevertheless, it is consistent with an ongoing degenerative process affecting *rho* edited rods. Importantly, this phenotype was observed in 81% (96/118) of crispant tadpoles that received a 500 pg dose, confirming the great efficiency of the designed sgRNA. In contrast, a lower dose of sgRNA (250 pg) did not significantly alter Rhodopsin staining (Figure 2E,F).

We then examined whether the loss of Rhodopsin expression might reflect rod cell death. Cleaved Caspase-3 protein labelling highlighted a significantly increased number of apoptotic cells in *rho* crispant neural retinas compared to control ones at all stages examined. This number was the highest by stage 40 (with a proportion of dying cells among rod photoreceptors estimated at 11.8% ± 2.2, *n* = 9) and declined thereafter (Figure 3A,B). As expected, co-labelling with Rhodopsin confirmed the rod identity of these dying cells (Figure 3A). In order to assess the extent of rod cell loss, we quantified the total number of nuclei in the outer nuclear layer, which in *Xenopus*, contains rods and cones with an approximate ratio of 1:1. It was significantly decreased in stage 45/47 *rho* crispants, but only by 8% (Figure 3C). This is far from the 50% that would be expected if all rods were dying, suggesting a preservation of most of them despite the virtual absence of Rhodopsin staining. In order to confirm this finding, rods were labelled with another specific marker, Recoverin, which stains both their nucleus and inner segment. Many Recoverin-positive cells were detected in stage 45 *X. tropicalis rho* crispants (Figure 3D), supporting the idea that rods are not massively dying. The staining area was however significantly reduced compared to controls, in part due to a lower thickness of the labelled layer (Figure 3D,E). This is suggestive of structural defects and consistent with the reduced expression of Rhodopsin (Figure 2E,F).

Finally, we aimed at assessing whether cone photoreceptors might degenerate secondarily to rod cells, as observed in humans. Immunostaining against Calbindin, a marker of cones in *Xenopus* [19], revealed a global conservation of these cells in stage 47 *rho* crispants (Figure 3F). However, the staining was thinner (Figure 3G), suggesting an abnormal morphology of outer segments. This could be confirmed at a later stage using the S/M opsin cone marker (Figure 3H).

As a whole, this *X. tropicalis rho* crispant model exhibits severe rod cell degeneration, including rod outer segment shortening and occasional cell death. This is likely accompanied by cone morphological defects as well, as suggested by the reduced expression of specific markers.

### 3.3. Characterization of Indels and Associated Phenotypes in F1 X. tropicalis rho Mutant Tadpoles

Retinal pathologies resulting from *Rho* mutations can be inherited in either an autosomal dominant or autosomal recessive manner. In order to obtain further insights into the genotype–phenotype link of our *rho* crispants, we generated F1 *X. tropicalis* tadpoles by crossing *rho* crispants with wild-type individuals. As expected, tide analysis of PCR products amplified from genomic DNA [14] revealed the presence of a wild-type allele and a single mutated one in each F1 tadpole (Figure 4C and data not shown). Among 29 F1 individuals analysed, DNA sequencing revealed 20 tadpoles with mutations at the level of the PAM site, with either reading frame-conserving or reading frame-shifting indels (Figure 4A). Only one tadpole with a three-base deletion (CAG), leading to the loss of Glutamine 28, showed altered Rhodopsin expression (Figure 4B,C). Of note, Glutamine 28 is a conserved amino acid (Figure 4D) whose mutation Q28H was previously described in patients with autosomal dominant retinitis pigmentosa [20]. All other tadpoles did not exhibit any significant modification of Rhodopsin labelling (Figure 4B and data not shown), suggesting that their mutations are either neutral or recessive. Altogether, these results suggest that the progressive degeneration of rod cells observed in mosaic F0 animals mainly results from the presence of two edited *rho* recessive alleles per cell (presumably leading to an absence of Rhodopsin expression or to the expression of a non-functional Rhodopsin protein) and more rarely from dominant mutations.

### 3.4. CRISPR/Cas9-Mediated Rhodopsin Editing in X. laevis

Having succeeded in editing the *rho* gene in *X. tropicalis*, we next applied the same strategy in *X. laevis* in order to compare the retinal response of both species to rod cell degeneration. Due to its pseudoallotetraploid nature, *X. laevis* genome possesses three genes encoding Rhodopsin, all located on chromosome 4 [11]: rho.S (XB-GENE-17342665), and two paralogous gene sequences, rho.L (XB-GENE-966893), and rho.2.L (XB-GENE-18034123). Importantly, the *rho* sgRNA designed for *X. tropicalis* could be used, as it showed 100% similarity with the target sequences present in the three *X. laevis rho* genes (Appendix A). Injection of 500 pg sgRNA together with 5 ng Cas9 protein was performed in *X. laevis* eggs. Expectedly [18], resultant F0 embryos showed mosaicism, as inferred from Tide analysis of PCR products amplified from genomic DNA (Appendix A). Sequencing of individual clones (*rho* sequences amplified from either single or pooled embryos) revealed that indels were generated with an efficacy of 94% (15/16, Appendix A).

As observed in *X. tropicalis*, H&E staining of *X. laevis* crispant retinas highlighted global preservation of nuclear layers but severe shortening of photoreceptor outer segments (Figure 5A). Immunostaining analysis further revealed the progressive disappearance of Rhodopsin staining within the central retina (Figure 5B), with only rare positive cells remaining at older stages (Figure 5C). Nearly 82% (147 among 178) of tadpoles presented this phenotype, showing *rho* gene-editing efficacy similar to that obtained in *X. tropicalis*. Apoptotic rods were found in these individuals at all analysed stages, with a peak by the end of embryogenesis (stage 39–41, proportion of dying cells among rod photoreceptors: 29.6% ± 6.3, *n* = 11). Their number then declined along with tadpole development (Figure 5D,E). However, similar to the *X. tropicalis* situation, the majority of rods survived in *X. laevis rho* crispants, as inferred from the quantification of outer layer nuclei and from the reduced but still visible Recoverin labelling (Figure 5F–H). Finally, analysis of S/M-opsin and calbindin expression highlighted that although cones were present in the whole outer nuclear layer of *X. laevis* crispants, their outer segment length was likely reduced (Figure 5I–K).

Altogether, these results highlight that the phenotype of *X. laevis* crispant retinas mimics the one observed in *X. tropicalis*, with severe rod cell degeneration, followed by the appearance of cone defects.

### 3.5. Comparison of X. laevis and X. tropicalis Müller Cell Response to Rod Degeneration

We previously demonstrated that *X. laevis* Müller cells actively re-enter into the cell-cycle and regenerate photoreceptors following mechanical lesion or selective ablation of rod photoreceptors [3]. We thus first wondered whether a similar regenerative response might occur in *X. laevis rho* crispants. Since we reported that Müller cell recruitment is more efficient in pre-metamorphic tadpoles [3], we focused our analysis on stage 56–60 individuals. A BrdU labelling assay revealed a 17-fold increase in proliferative cells within the central retina of *X. laevis* crispant tadpoles compared to controls (Figure 6A–C). Further analysis revealed that a significant proportion of PCNA-positive cycling cells were Müller cells, as assessed by co-staining against YAP (Figure 6D,E) or SOX9 (Appendix A), two specific markers [21,22]. This result thus shows that, despite the limited extent of rod cell death in this CRISPR-Cas9 model, their degeneration nonetheless triggers Müller glia proliferative responses in pre-metamorphic tadpoles.

In such a CRISPR-Cas9 model where *rod* mutations are inherited within cell lineages, evaluation of the regeneration process, if it occurs, is complicated by the fact that lost cells will be replaced by cells susceptible to degenerate as well. In line with this idea, we did not observe full or partial rod regeneration in crispant *X. laevis* retinas. However, we occasionally found BrdU–Rhodopsin co-labelled cells (Figure 6B), which might represent newborn rods. It is thus likely that Müller cells are attempting to regenerate lost photoreceptors in this *rho* crispant model as they do upon mechanical lesion or selective photoreceptor ablation.

A similar analysis of cell proliferation was then performed in *X. tropicalis rho* crispants (Figure 6F,G). Although the number of BrdU-positive cells was significantly increased compared to controls, it remained quite anecdotic, being 20 times lower than that observed in *X. laevis* crispant retinas (about 5 versus 100 cells per section, comparing Figure 6B,C,F,G). Of note, an important and significant difference was still observed when the number of proliferative cells was normalized to retinal size (Figure 6H). These observations reveal that *X. tropicalis* Müller glial cells are less prone than *X. laevis* ones to re-enter into the cell cycle upon rod degeneration.

As mentioned in the introduction, the CMZ constitutes another cellular source for retinal self-repair. This neurogenic region is particularly crucial for regeneration in *X. tropicalis* retinectomized animals, which in contrast to *X. laevis* ones, do not use RPE as a cellular source [9]. We therefore asked whether potential attempts of *X. tropicalis* crispants to replace lost rods might rely on CMZ cells rather than on Müller glia. Clearly, in mutant retinas from both species, constant rod production is maintained. Despite the virtual absence of Rhodopsin-labelled cells in the central retina, a distinct staining is observable in direct continuity with the CMZ (Figure 7A,B). This progressively fades along the peripheral to central axis, most probably reflecting cycles of rod genesis and subsequent degeneration. We found an increased number of BrdU-labelled cells in *X. tropicalis* crispant retinas compared to controls. This suggests an attempt to regenerate lost rods through CMZ cell mobilization. In contrast, such increased peripheral proliferation was not detected in *X. laevis* crispants (Figure 7C).

Altogether, these data indicate that both CMZ and Müller cells respond differently in *X. laevis* and *X. tropicalis* upon photoreceptor degeneration, highlighting divergent mechanisms of retinal regeneration in these two amphibian models.

## 4. Discussion

CRISPR-Cas9 technology serves as an efficient tool for genome editing and has opened new avenues for human disease modelling in *Xenopus* [23]. We took advantage of this approach to generate both *X. laevis* and *X. tropicalis* models of retinitis pigmentosa, a genetic condition that leads to progressive and inescapable vision loss. In both species, crispant individuals exhibit extensive rod degeneration. However, Müller glia response to this pathological environment differs tremendously between the two species (Figure 8). While these glial cells actively re-enter into the cell cycle in *X. laevis*, their proliferative response remains highly limited in *X. tropicalis*, which likely mobilizes CMZ cells. Altogether, our study thus reveals divergent evolution of regenerative mechanisms in these two closely related species.

We previously implemented two retinal injury paradigms in *X. laevis*: a mechanical needle poke injury and a transgenic model based on the Nitroreductase/Metronidazole system, allowing for conditional rod ablation [3]. Although pertinent to study the mechanisms underlying Müller cell-dependent regeneration, they do not mimic the situation of chronic degenerative diseases found in human patients. We thus turned to a CRISPR/Cas9 model, using a sgRNA targeting the *rho* gene. A similar strategy was previously employed in *X. laevis* to generate knockout, dominant alleles, and targeted insertions in this gene [11]. In this study, *rho* crispants were reported to exhibit missing and malformed rod photoreceptors resulting from both recessive and dominant mutations in the *rho* gene. Our data confirm these findings and further show a worsening of this degenerative phenotype along with age, as well as the occurrence of secondary cone defects. These could either be due to the narrowing of the space separating photoreceptors from the RPE (due to the absence or malformed rod outer segments) or may arise as a lack of neuroprotective rod signals. Such secondary alteration of cones is highly reminiscent of the situation observed in human patients with retinitis pigmentosa. Given the high cone/rod ratio of *Xenopus* (about 50% of cones, compared to 3–9% in the mouse), *rho Xenopus* crispants offer an attractive model to develop cone preservation strategies in a rod degenerative context, a highly relevant topic in ophthalmology. Of note, we found that although malformed, rods were not massively dying in our *Xenopus* models, with a loss of cells limited to 8–20%. This contrasts with *rho* knockout rods in mice that undergo extensive cell death, although with a variable extent depending on the genetic context [24,25]. The high rod survival rate observed in *Xenopus* at tadpole stages suggests the existence of neuroprotective mechanisms, which would be worth investigating.

Retinal self-repair abilities tremendously vary among vertebrates both in terms of efficiency and cellular sources involved. We here report that *X. laevis* and *X. tropicalis* Müller cells differentially respond to photoreceptor degeneration. This finding further exemplifies the divergences that exist among amphibian species with regards to regenerative mechanisms. Following retinectomy for instance, *X. tropicalis* exploit CMZ cells to reconstitute the neural retina, while this mainly occurs upon RPE reprogramming in newt and *X. laevis* [9,26]. In addition, in these two species, differences were identified regarding the mechanisms underlying RPE-dependent regeneration [8,27]. Thus far, the molecular cues that restrict or trigger the recruitment of a particular cellular source upon injury largely remain to be identified. Inter-species comparisons may help in this way, as exemplified by a recent report from Wittbrodt lab. In their study, the authors show that medaka Müller cells differ from zebrafish ones, by exhibiting a limited regenerative capacity [28]. Their potential can however be upgraded by sustaining the expression of Sox2, a transcription factor involved in Müller glia self-renewal [28]. Intrinsic factors might explain as well the limited proliferative potential of *X. tropicalis* Müller cells in *rho* crispants. Alternatively, extrinsic signals might be involved. Nutrient availability was recently proposed as a contributor of tail regenerative competence in *X. tropicalis* [29]. In this species, as previously observed in *X. laevis*, the authors identified a refractory period where tail regeneration is hampered and showed that it can be circumvented by alleviating nutrient stress. It would thus be interesting to assess whether *X. tropicalis* Müller glia would be more prone to proliferate if tadpoles were fed more abundantly or differently than usual. The limited recruitment of Müller cells in *X. tropicalis rho* crispants could also result from insufficient signals emanating from damaged cells. In this regard, the global sparing of cone cells might play a role. It was shown that zebrafish mutants with selective rod or cone degeneration display distinct proliferative responses, being restricted to rod precursors in the first case, while relying on Müller glia for the second [30]. Similarly, only outer layer progenitors were found to proliferate in a zebrafish transgenic model of retinitis pigmentosa carrying the P23H mutation [31]. It would thus be useful to address whether a different injury paradigm that would cause massive cone apoptosis may trigger efficient *X. tropicalis* Müller glia proliferation.

In conclusion, the divergent efficacy of Müller glia response to injury that we identified in *X. laevis* and *X. tropicalis* offers the possibility to uncover the molecular cues accounting for such variability. This opens new avenues to understand the mechanisms that have shaped regeneration during evolution, with tremendous differences across vertebrates and even among closely related species.

## Figures and Tables

**Figure 1 cells-11-00807-f001:**
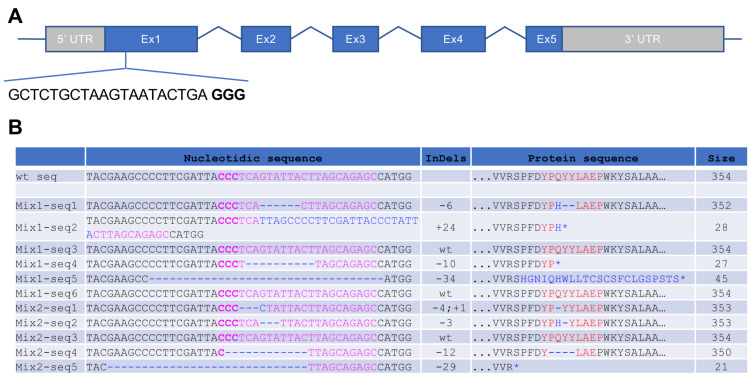
Genomic analysis of *X. tropicalis* rho crispants. (**A**) Schematic of the *Xenopus tropicalis rho* gene. The depicted sequence corresponds to the CRISPR RNA targeting exon 1 and to the protospacer adjacent motif (PAM; indicated in bold). (**B**) Sequences obtained from individual clones of PCR products amplified from genomic DNA. Two mixtures (Mix1, 2) of 5 *X. tropicalis* F0 embryos injected with 500 pg of *rho* sgRNA and 600 pg of *cas9* mRNA were used. For each clone, the targeted sequence is shown in pink, with PAM in bold, while indels are in blue. The right column indicates the corresponding protein sequence and its size (number of amino acids).

**Figure 2 cells-11-00807-f002:**
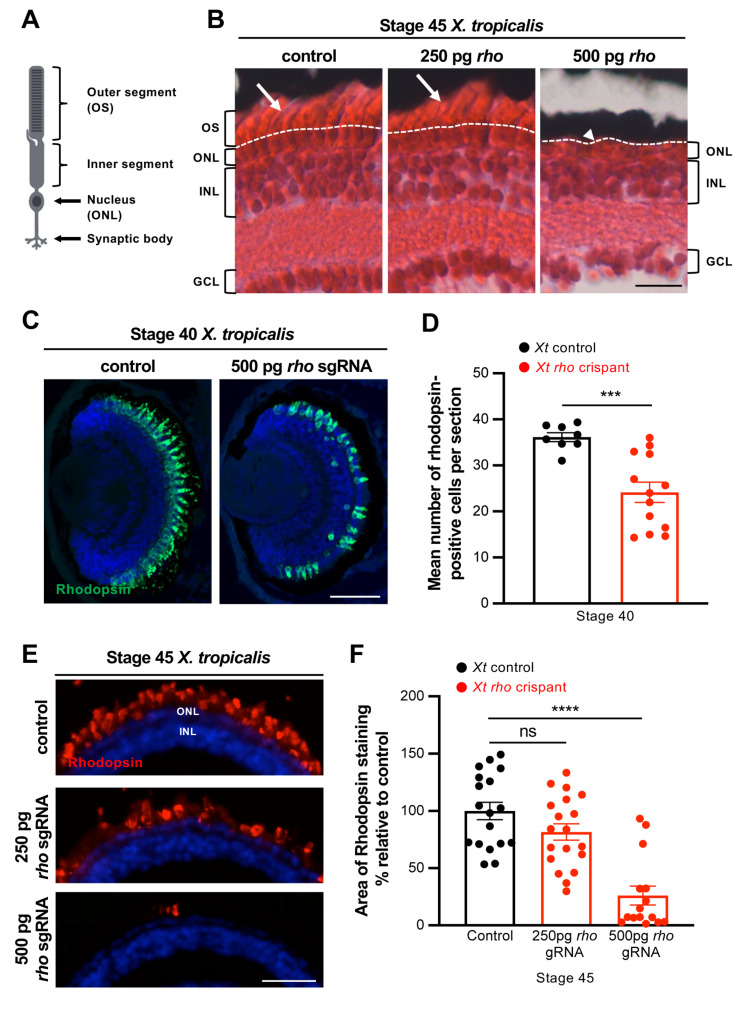
Rod degeneration in *X. tropicalis rho* crispants. (**A**) Schematic representation of a rod photoreceptor. (**B**) Hematoxylin and Eosin staining on retinal sections from stage 45 control and *rho* crispant *X. tropicalis* tadpoles. The dotted line delineates the border between the outer nuclear layer and photoreceptor outer segments (arrows). The arrowhead indicates the absence of outer segments. (**C**–**F**) Immunofluorescence analysis of Rhodopsin expression on retinal sections from stage 40 (**C**,**D**) and stage 45 (**E**,**F**) control and crispant *X. tropicalis* tadpoles. Rhodopsin labels the rod outer segments. The number of Rhodopsin-positive cells and the area of Rhodopsin staining are quantified in (**D**,**F**), respectively. In (**C**,**E**), cell nuclei are counterstained with Hoechst (blue). In graphs, data are represented as mean ± SEM and each point represents one retina. *** *p* < 0.001; **** *p* < 0.0001; ns: non-significant (Mann–Whitney tests). GCL: ganglion cell layer, INL: inner nuclear layer, ONL: outer nuclear layer, OS: outer segment. Scale bars: 25 µm in (**B**,**E**) and 50 µm in (**C**).

**Figure 3 cells-11-00807-f003:**
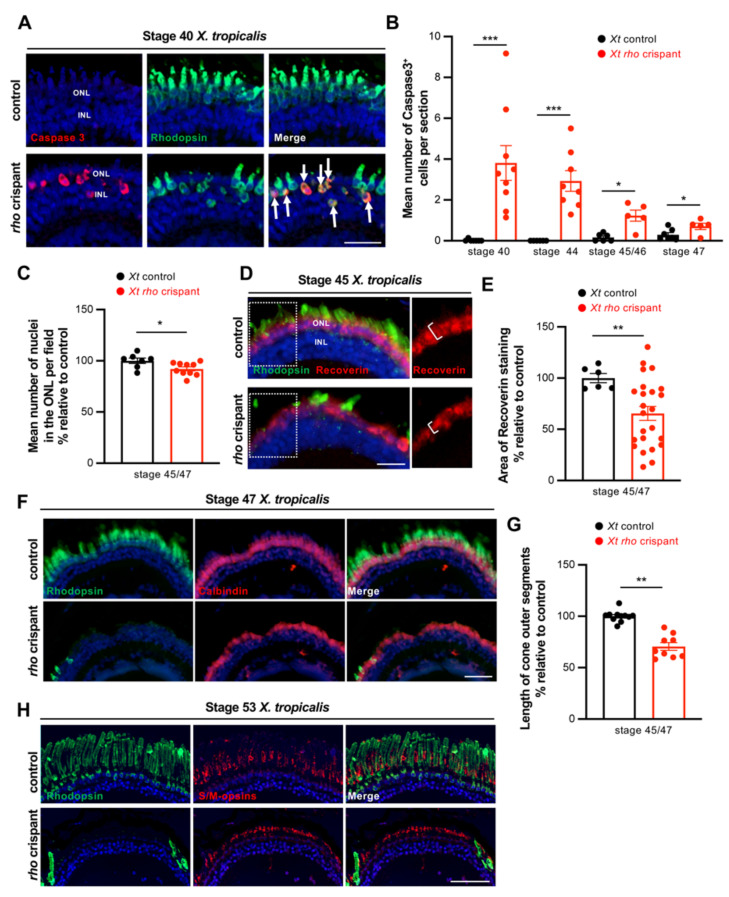
Rod cell death and cone defects in *X. tropicalis rho* crispants. (**A**) Typical retinal sections from stage 40 control and crispant *X. tropicalis* embryos, immunostained for cleaved Caspase 3 and Rhodopsin. Arrows point to double positive cells. (**B**) Quantification of Caspase 3-positive cells from stage 40 to stage 47. (**C**) Quantification of ONL nuclei on retinal sections from stage 45/47 control and crispant *X. tropicalis* tadpoles. Nuclei were counted in one standard square field per section. (**D**,**E**) Immunofluorescence analysis of Recoverin expression on retinal sections from stage 45 control and crispant *X. tropicalis* tadpoles. Sections were co-stained for Rhodopsin for comparison. Panels on the right show the regions delineated with the dotted white boxes and highlight the decreased thickness of Recoverin staining in crispants (brackets). The total area of Recoverin staining is quantified in (**E**). (**F**) Typical retinal sections from stage 47 control and crispant *X. tropicalis* tadpoles, immunostained for Rhodopsin and Calbindin (a marker of cone photoreceptors). (**G**) Quantification of cone outer segment length. (**H**) Retinal sections from stage 53 control and crispant *X. tropicalis* tadpoles, immunostained for Rhodopsin and S/M-opsin (a marker of cone photoreceptors). Note the continuous but thinner staining in the outer nuclear layer of crispant tadpoles. In (**A**,**D**,**F**,**H**), cell nuclei are counterstained with Hoechst (blue). In graphs, data are represented as mean ± SEM, and each point represents one retina. * *p* < 0.05; ** *p* < 0.01; *** *p* < 0.001 (Mann–Whitney tests). Scale bar: 25 µm in (**A**,**D**) and 50 µm in (**F**,**H**).

**Figure 4 cells-11-00807-f004:**
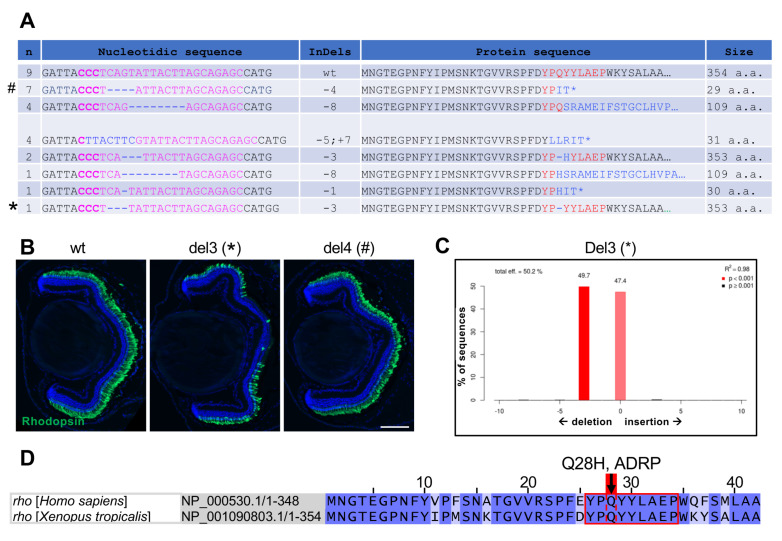
Characterization of F1 *X. tropicalis rho* mutants. (**A**) Genomic sequences retrieved from F1 *X. tropicalis rho* mutants. The left column indicates the number of tadpoles harbouring the corresponding sequence. The targeted region is shown in pink with PAM in bold, while indels are in blue. The right column indicates the resulting protein sequence and its size (number of amino acids). (**B**) Immunofluorescence analysis of Rhodopsin expression on retinal sections from stage 45 wild-type and F1 individuals. Del3 and del4 tadpoles bear the sequences depicted in A with * (leading to lack of glutamine 28) and # (leading to a truncated protein), respectively. Cell nuclei are counterstained with Hoechst (blue). (**C**) TIDE analysis of the del3 *rho* mutant showing the presence of a wild-type and a mutated allele. (**D**) Illustration of *Homo sapiens* and *X. tropicalis* Rhodopsin protein sequences (amino acids 1 to 42). The region targeted by the *rho* sgRNA (red square) is conserved between the two species and includes glutamine 28, whose mutation Q28H was found in patients with autosomal dominant retinitis pigmentosa (ADRP). Scale bar: 100 µm.

**Figure 5 cells-11-00807-f005:**
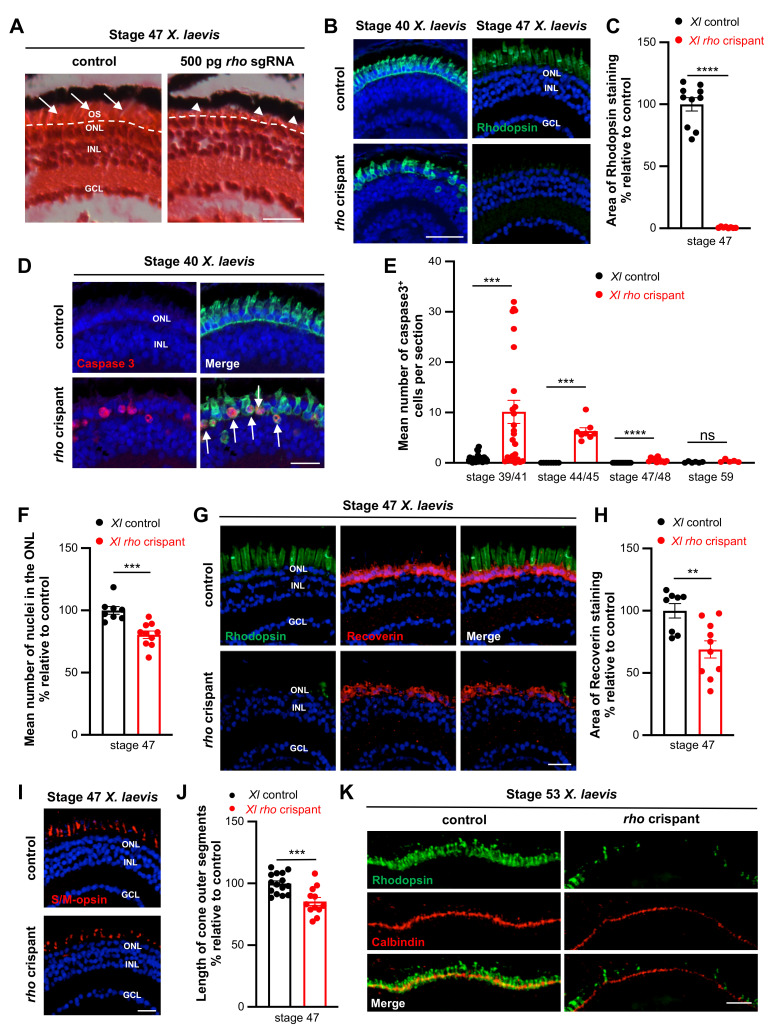
Rod degeneration and cone defects in *X. laevis rho* crispants. (**A**) Hematoxylin and Eosin staining on retinal sections from stage 47 control and *rho* crispant *X. laevis* tadpoles. The dotted line delineates the border between the outer nuclear layer and photoreceptor outer segments (arrows). Arrowheads point to shortened outer segments. (**B**,**C**) Immunofluorescence analysis of Rhodopsin expression on retinal sections from control and crispant *X. laevis* tadpoles at stage 40 or 47. The area of Rhodopsin staining at stage 47 is quantified in (**C**). (**D**) Typical retinal sections from stage 40 control and crispant *X. laevis* embryos, immunostained for cleaved Caspase 3 and Rhodopsin. Arrows point to double positive cells. (**E**) Quantification of Caspase 3-positive cells from stage 39/41 to stage 59. (**F**) Quantification of ONL nuclei on retinal sections from stage 47 control and crispant *X. laevis* tadpoles. Nuclei were counted in one standard rectangle field per section. (**G**,**H**) Immunofluorescence analysis of Recoverin expression on retinal sections from stage 47 control and crispant *X. laevis* tadpoles. Sections were co-stained for Rhodopsin for comparison. The area of Recoverin staining is quantified in (**H**). (**I**,**J**) Immunofluorescence analysis of S/M-opsin expression on stage 47 control and crispant *X. laevis* tadpoles. The length of S/M opsin-labelled outer segments is quantified in (**J**). (**K**) Typical retinal sections from stage 53 control and crispant *X. laevis* tadpoles, immunostained for calbindin. Note the continuous but thinner staining in the outer nuclear layer of crispant tadpoles. In (**B**,**D**,**G**,**I**), cell nuclei are counterstained with Hoechst (blue). In graphs, data are represented as mean ± SEM and each point represents one retina. ** *p* < 0.01; *** *p* < 0.001; **** *p* < 0.0001; ns: non-significant (Mann–Whitney tests). GCL: ganglion cell layer, INL: inner nuclear layer, ONL: outer nuclear layer, OS: outer segment. Scale bar: 25 µm in (**A**,**B**,**D**,**H**,**I**) and 50 µm in (**K**).

**Figure 6 cells-11-00807-f006:**
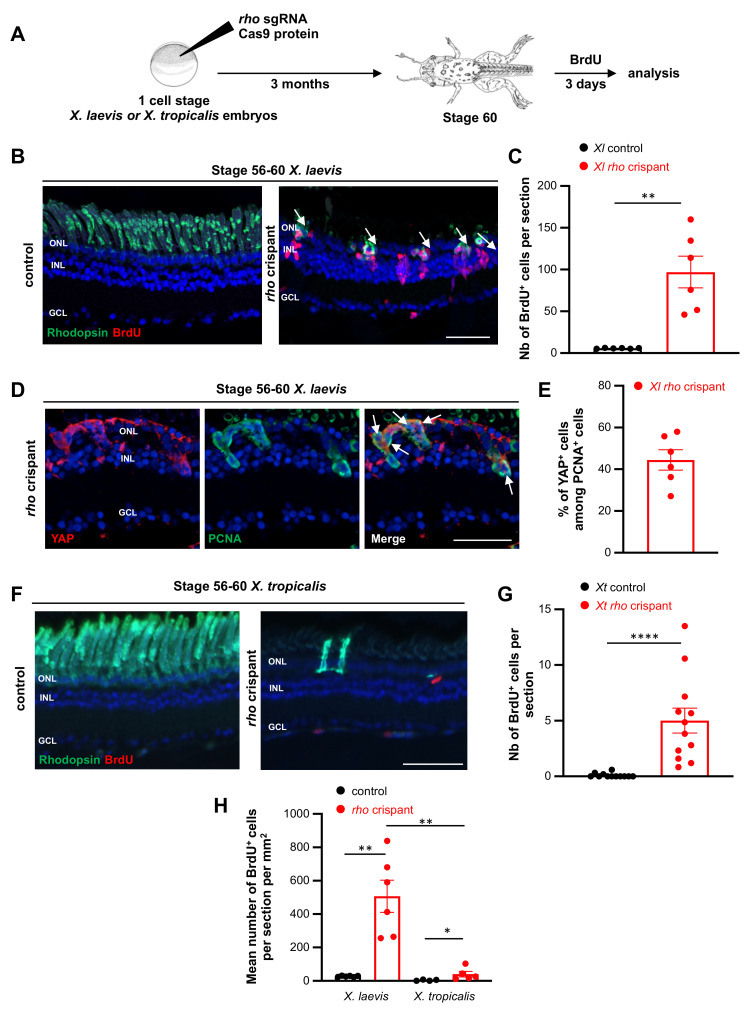
Analysis of proliferation in *X. tropicalis* and *X. laevis rho* crispants. (**A**) Timeline diagram of the experimental procedure used in (**B**–**H**). (**B**,**C**) BrdU assay on retinal sections from stage 56–60 control and crispant *X. laevis* tadpoles. Sections were co-labelled for Rhodopsin. Arrows point to double positive cells. BrdU-positive cells are quantified in (**C**). (**D**) Typical sections from stage 56–60 crispant *X. laevis* tadpoles, immunostained for PCNA (a marker of proliferative cells) and YAP (a marker of Müller cells). Arrows point to double positive cells. (**E**) Quantification of proliferating Müller cells (YAP- and PCNA-positive). (**F**,**G**) BrdU assay on retinal sections from stage 56–60 control and crispant *X. tropicalis* tadpoles. Sections were co-labelled for Rhodopsin. BrdU-positive cells are quantified in (**G**). (**H**) Comparison of BrdU incorporation between *X. laevis* and *X. tropicalis* crispants at stage 56–60. The number of BrdU-positive cells was normalized to retinal cell surface. In (**B**,**D**,**F**) cell nuclei are counterstained with Hoechst (blue). In (**C**,**G**), counting was performed in the central retina, considering only the inner and outer nuclear layers. In graphs, data are represented as mean ± SEM and each point represents one retina. * *p* < 0.05; ** *p* < 0.01; **** *p* < 0.0001 (Mann–Whitney tests). GCL: ganglion cell layer, INL: inner nuclear layer, ONL: outer nuclear layer. Scale bars: 50 µm in (**B**,**F**), 25 µm in (**D**).

**Figure 7 cells-11-00807-f007:**
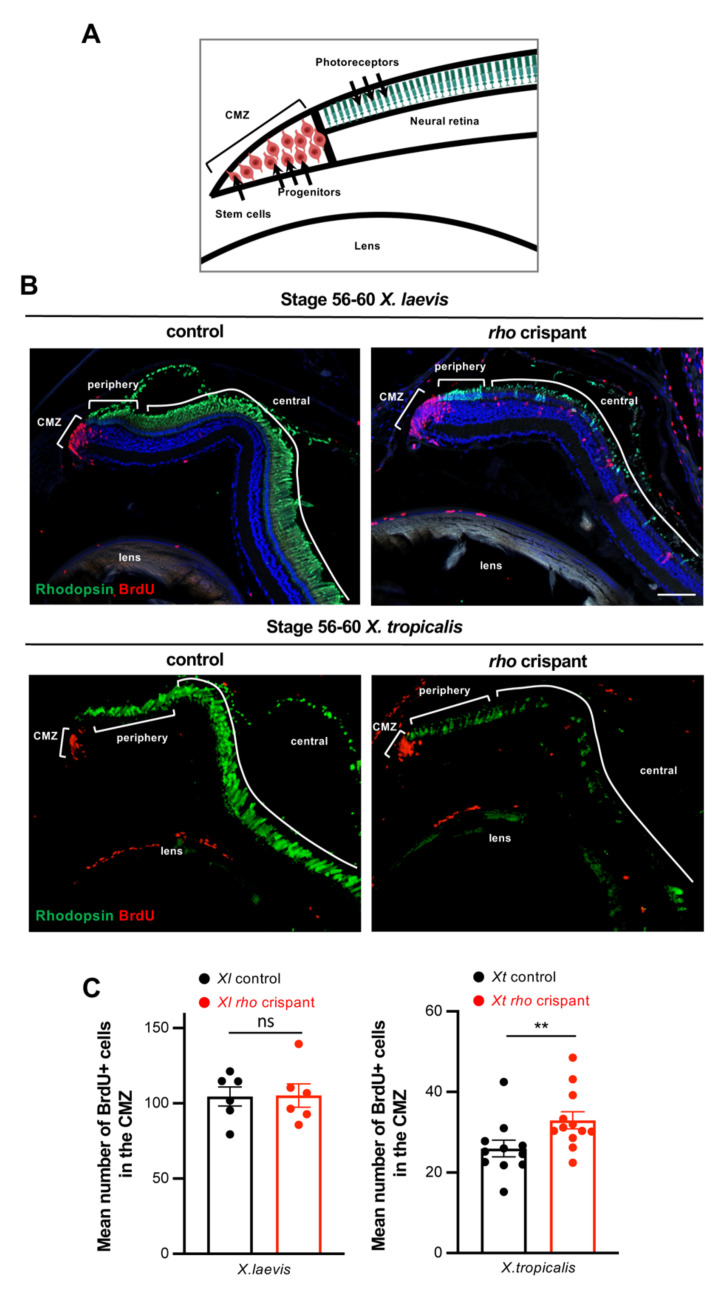
Analysis of proliferative CMZ cells in *X. tropicalis* and *X. laevis rho* crispants. (**A**) Schematic representation of the peripheral part of the retina. Stem cells and progenitors lie at the tip within the ciliary marginal zone (CMZ), while newly produced differentiated cells are found more centrally (only photoreceptors are shown here). (**B**) Typical retinal sections from stage 60 control and crispant *X. laevis and X. tropicalis* tadpoles, immunostained for Rhodopsin and BrdU (protocol illustrated in the timeline diagram of Figure 6A). Cell nuclei are counterstained with Hoechst (blue). (**C**) Quantification of BrdU-labelled cells in the CMZ. In graphs, data are represented as mean ± SEM, and each point represents one retina. ** *p* < 0.01; ns: non-significant (Mann–Whitney tests). Scale bar: 50 µm.

**Figure 8 cells-11-00807-f008:**
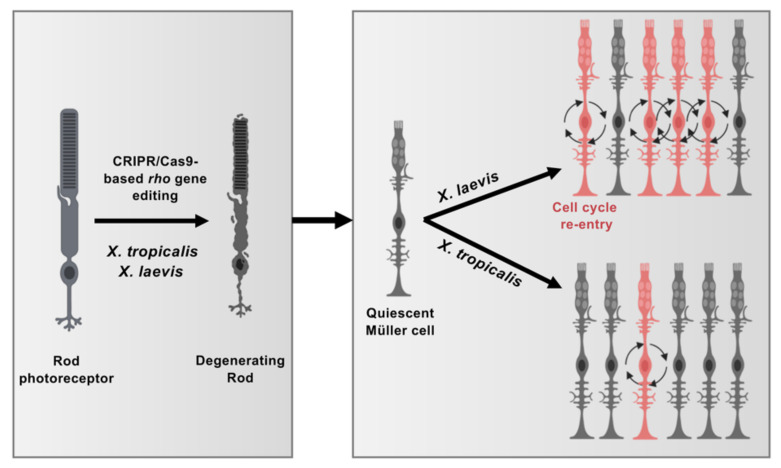
Model illustrating differences in Müller cell response to injury in *X. tropicalis* and *X. laevis*. CRISPR/Cas9-mediated *rho* editing triggers rod photoreceptor degeneration in both *X. tropicalis* and *X. laevis*. Quiescent Müller cells respond to this pathological context by re-entering into cell cycle. While Müller cell proliferation is greatly enhanced in *X. laevis*, it remains limited in *X. tropicalis*, suggesting divergent regenerative properties of these closely related species. This figure was created with schemas from ©BioRender—biorender.com accessed on November 2021.

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
