# Peer review of "CRISPR/Cas9-Mediated Models of Retinitis Pigmentosa Reveal Differential Proliferative Response of Müller Cells between Xenopus laevis and Xenopus tropicalis"

_cells, 2022, doi:10.3390/cells11050807_

Round 1

Reviewer 1 Report

Authors used CRISPR/Cas9 technology to develope rhodopsin gene editing-based models of retinitis pigmentosa in two Xenopus species, Xenopus laevis and Xenopus tropicalis. The model was proved to be successful. Their unexpected finding was that Müller glial cells behave very differently in Xenopus laevis and Xenopus tropicalis. This work will be very interesting to Muller biology community. The work will shed light on our understanding of the mechanisms involved in retinal regeneration .

Author Response

Dear Reviewer 1,

We would like to thank you, Reviewer 1, for your time and positive comment.

Sincerely,

Morgane Locker and Muriel Perron

Reviewer 2 Report

This manuscript by Parain et al. describes the use of CRISPR/Cas9 to edit rhodopsin expression in two species of Xenopus as a mechanism to generate inherited retinal dystrophy models of retinitis pigmentosa. This approach has been used previously in other labs (e.g., Feehan et al. 2017) in Xenopus laevis, so this aspect of the paper is not unique, but is rather confirmatory. What is unique in this paper is that the authors identify that the two species of Xenopus respond uniquely to the rod damage, with X. laevis initiating a strong Muller glia response, whereas X. tropicalis primarily responds at the ciliary margin. This finding is interesting and lays the groundwork for future studies of the differences in the cellular responses in these two closely related species. These studies will likely be very informative in defining the pathways that lead to regeneration of photoreceptors and thus have significant impact.

Overall, the studies are skillfully conducted with good controls, and the results are clearly presented. Consequently, I have only a few minor criticisms that should be addressed in a revision.

Comments:

  • In Fig 2D and 2F, there are clear outliers that largely do not show a response to the sgRNA in rhodopsin expression. Do the authors have any data on the nature of these outliers? Is there a correlation of individual TIDE with changes in rhodopsin, such that these are simply individuals in which gene editing just didn’t occur?
  • P7, line 218: The authors indicate the staining is “significantly affected”. Please describe what the authors are seeing, since I do not detect an appreciable difference.
  • P7, line 227: the authors conclude that there is “altered cone morphology.” This conclusion reaches beyond the images presented. The best diagnostic would be electron microscopic images, but perhaps the authors could attain sufficient detail with higher magnification (~100X) confocal imaging of the cones since Xenopus cones are pretty large.
  • Fig 3B: This figure panel would be informative and diagnostic if presented as “percent of rod photoreceptors”, particularly in light of Fig 3D which shows that most rod cell bodies are retained.
  • Fig 3H: Something is wrong with either the rhodopsin antibody or the secondary antibody since both rods and cones are stained (this is particularly evident in the rho crispants where the rod outer segments are gone, but can also be seen in the controls).
  • P6, line 298: The authors conclude an “altered morphology” for the laevis crispants in Fig 5F-H. As in comment #3 above, this is very difficult (impossible?) to discern at the level of magnification presented. Again, higher magnification confocal microscopy or EM would provide more clear resolution.
  • P14, Fig 3S: This figure is an essential part of the manuscript and should not be relegated to the supplement. It should be moved to the main body.

Author Response

Dear Reviewer 2,

We would like to thank you, Reviewer 2, for your time and your suggestions. Please find below our point-by-point responses to your comments.

Sincerely,

Morgane Locker and Muriel Perron

  • In Fig 2D and 2F, there are clear outliers that largely do not show a response to the sgRNA in rhodopsin expression. Do the authors have any data on the nature of these outliers? Is there a correlation of individual TIDE with changes in rhodopsin, such that these are simply individuals in which gene editing just didn’t occur?

We do not know the nature of the outliers since in this experiment the TIDE analysis was done on different individuals than the ones used for Rhodopsin expression analysis. In each set of experiments, we always found a few tadpoles that do not exhibit degeneration, probably indeed because Cas9 editing did not occur.

  • P7, line 218: The authors indicate the staining is “significantly affected”. Please describe what the authors are seeing, since I do not detect an appreciable difference.

Recoverin expression was quantified by measuring the staining area. We found that the total labelled surface was decreased in crispants compared to controls, in part due to a lowering of the staining thickness. To make this clear, we added panels showing only Recoverin staining in Fig. 3D, modified the legend accordingly, and rewrote our sentence as follows: “The staining area was however significantly reduced compared to controls, in part due to a lower thickness of the labelled layer (Figure 3D, E). This is suggestive of structural defects and consistent with the reduced expression of Rhodopsin (Figure 2F).” We also added in the material & methods section a more detailed explanation of the quantification.

  • P7, line 227: the authors conclude that there is “altered cone morphology.” This conclusion reaches beyond the images presented. The best diagnostic would be electron microscopic images, but perhaps the authors could attain sufficient detail with higher magnification (~100X) confocal imaging of the cones since Xenopus cones are pretty large.

We agree that such “altered morphology” conclusion would require to be supported by electron microscopy analyses. We thus tuned down our conclusion and made it clear that the altered cone morphology is a suggestion based on the reduced cone marker expression: “As a whole, this X. tropicalis rho crispant model exhibits severe rod cell degeneration, including rod outer segment shortening and occasional cell death. This is likely accompanied by cone morphological defects as well, as suggested by the reduced expression of specific markers.”

  • Fig 3B: This figure panel would be informative and diagnostic if presented as “percent of rod photoreceptors”, particularly in light of Fig 3D which shows that most rod cell bodies are retained.

We agree that providing the percentage of dying photoreceptors in addition to the number of dying cells per section would be informative. We have added this in the text as follows: “This number was the highest by stage 40 (with a proportion of dying cells among rod photoreceptors estimated at 11.8% ± 2.2, n=9) and declined thereafter (Figure 3A, B).

We also have also added the same measurement for X. laevis: “Apoptotic rods were found in these individuals at all analysed stages, with a peak by the end of embryogenesis (stage 39-41; proportion of dying cells among rod photoreceptors: 29.6% ± 6.3, n=11). Their number then declined along with tadpole development (Figure 5D, E). ”

  • Fig 3H: Something is wrong with either the rhodopsin antibody or the secondary antibody since both rods and cones are stained (this is particularly evident in the rho crispants where the rod outer segments are gone, but can also be seen in the controls).

We thank the reviewer for having noticed this. We indeed agree that on these pictures, the Rhodopsin labelling gave some background staining in cones, probably due to typical auto-fluorescence. We thus took other pictures from the very same experiment, with less exposure time to reduce background. We have now replaced panels in Fig. 3H.

  • P6, line 298: The authors conclude an “altered morphology” for the laevis crispants in Fig 5F-H. As in comment #3 above, this is very difficult (impossible?) to discern at the level of magnification presented. Again, higher magnification confocal microscopy or EM would provide more clear resolution.

As for comment #3, we agree that our “altered morphology” conclusion would require to be supported by electron microscopy analysis. We thus modified the sentence as follows: “However, similarly to the X. tropicalis situation, the majority of rods survived in X. laevis rho crispants, as inferred from the quantification of outer layer nuclei and by the reduced but still visible Recoverin labelling (Figure 5F-H).”

We also accordingly modified the two following sentences: “Finally, analysis of S/M-opsin and calbindin expression highlighted that although cones were present in the whole outer nuclear layer of X. laevis crispants, their outer segment length was likely reduced (Figure 5I-K).” “Altogether, these results highlight that the phenotype of X. laevis crispant retinas mimics the one observed in X. tropicalis, with severe rod cell degeneration, followed by the appearance of cone defects”.

  • P14, Fig 3S: This figure is an essential part of the manuscript and should not be relegated to the supplement. It should be moved to the main body.

We agree and have now moved this supplementary figure S3 as Figure 7.

Reviewer 3 Report

This manuscript by Parain et al. describes the generation of models for retinitis pigmentosa in two related Xenopus species using CRISPR-mediated mosaic knockouts in the rho gene. Remarkably, while the retinal defects in the two species are very similar, the regenerative responses were very different. While for X. laevis Muller cell proliferation was induced, mirroring the situation in zebrafish, this was not the case for X. tropicalis. The differential response in two very related species opens the possibility for future identification of the intrinsic and/or extrinsic factors involved.

Overal this manuscript is well written and the data are clearly presented. All experiments are properly controled. I only have a few minor comments.

1) In paragraph 3.1. the authors mention that the indels detected upon sequencing would result in the expression of a truncated protein (in case of a frame-shift mutation). The more likely event is absence of an expressed protein due to non-sense mediated decay. Also, an in-frame indel will result in an amino acid deletion, instead of a point mutation (= the term used in the MS)

2) The resolution of the H&E staining in Fig. 2B is rather low making it difficult to see the histological defect. It may be better (if sections are still available) to check the use of toluidine blue to obtain better histological resolution.

3) The argument for a secondary effect on the cones is rather weak. The thinning of the outer segment layer (Fig. 3F and 3H) may be due to the narrowing of the space between the photoreceptors and the RPE (due to absence of malformed rod outer segments).

4) Section 3.3. final sentence, include the possible absence of rhodopsin expression (rather than loss-of-function) due to bi-allelic out-of-frame alleles.

5) line 283, 'fertilized eggs' instead of 'fertilized oocytes'

Author Response

Dear Reviewer 3,

We would like to thank you, Reviewer 3, for your time and your suggestions. Please find below our point-by-point responses to your comments.

Sincerely,

Morgane Locker and Muriel Perron

1) In paragraph 3.1. the authors mention that the indels detected upon sequencing would result in the expression of a truncated protein (in case of a frame-shift mutation). The more likely event is absence of an expressed protein due to non-sense mediated decay. Also, an in-frame indel will result in an amino acid deletion, instead of a point mutation (= the term used in the MS)

We have now modified the sentence as follows: “Among analysed sequences, 73% (8/11) exhibited insertion–deletion (indels). 4 of them led to frameshifts (probably resulting in truncated proteins or nonsense-mediated decay), and 4 others generated amino acid deletions (Figure 1B).”

2) The resolution of the H&E staining in Fig. 2B is rather low making it difficult to see the histological defect. It may be better (if sections are still available) to check the use of toluidine blue to obtain better histological resolution.

We have taken new pictures from the very same experiment. We believe that this time the severe effects on the outer segments are more obvious in this new Fig. 2B.

3) The argument for a secondary effect on the cones is rather weak. The thinning of the outer segment layer (Fig. 3F and 3H) may be due to the narrowing of the space between the photoreceptors and the RPE (due to absence of malformed rod outer segments).

We agree that cone defects could either be due to the lack of neuroprotective signals from rods that are degenerating or indirectly result from the narrowing of the space separating them from the RPE. We have thus added this hypothesis in our discussion: “Our data confirm these findings and further show a worsening of this degenerative phenotype along with age, as well as the occurrence of secondary cone defects. These could either be due to the narrowing of the space separating photoreceptors from the RPE (due to the absence or malformed rod outer segments) or may arise as a lack of neuroprotective rod signals. Such secondary alteration of cones is highly reminiscent of the situation observed in human patients with retinitis pigmentosa.”

4) Section 3.3. final sentence, include the possible absence of rhodopsin expression (rather than loss-of-function) due to bi-allelic out-of-frame alleles.

We have now modified our sentence as follows: “Altogether these results suggest that the progressive degeneration of rod cells observed in mosaic F0 animals mainly results from the presence of two edited rho recessive alleles per cell (presumably leading to an absence of Rhodopsin expression or to the expression of a non-functional Rhodopsin protein) and more rarely from dominant mutations.”

5) line 283, 'fertilized eggs' instead of 'fertilized oocytes'

We have replaced fertilized oocytes by eggs: “Injection of 500 pg sgRNA together with 5 ng Cas9 protein was performed in X. laevis eggs.”

Reviewer 4 Report

The work of Parain et al. introduces two new models of retinitis pigmentosa in two Xenopus species, which show a differential regeneration respond of the retina. They aim to provide novel insights into the function and activity of the Müller glial cells in this respect to better understand how regeneration of the retina is achieved but also restricted. Though the amount of novel data above what has been published before by the same laboratory is limited, the data is consistent and of very high quality. Moreover, it opens new roads to understand mechanistical differences between two close species and may help to improve the therapeutic intervention for retinal degenerative diseases in humans in the future.

In general, the technical quality and the presentation of the data is sound and is of interest for the readership of Cells after very minor revision.

Minor points:

In the introduction/discussion I do miss a little bit the human Müller glial cell (function in the adult human retina vs. larval Xenopus retina). Is there retinal regeneration of the embryonic retina in mice in utero?

Since newborn photoreceptor cell are defective in crispants as well, did the authors try to rescue the defect by the coinjection of “inducible” rhodopsin or lipofection/electroporation of expression plasmids?

Discussion: I wonder about the fact that in frogs and in fish the pseudotetraploid possesses a better regenerative capacity than the diploid counterpart. It there a kind of dose-effect of the respective gene, different epigenetic regulation? Moreover, I wonder whether it is possible to perform single cell analysis of Müller cells for comparison.

How much is known about endocytosis/autophagy/energy metabolism of the Müller glial cells (retinal cells) under regeneration? (AMPK, p62…, rab8)?

Typos:

Wording at the beginning of figure legend 3. …. (A) Typical retinal secTable 40. (?)

Wording line 5 of figure legend 5. …. Scheme 40. or 47. (?)

Author Response

Dear Reviewer 4,

We would like to thank you, Reviewer 4, for your time and your suggestions. Please find below our point-by-point responses to your comments.

Sincerely,

Morgane Locker and Muriel Perron

In the introduction/discussion I do miss a little bit the human Müller glial cell (function in the adult human retina vs. larval Xenopus retina). Is there retinal regeneration of the embryonic retina in mice in utero?

We have now added a few sentences in the introduction section to provide more information about the normal function of Müller cells and their regenerative potential in mammals:

“Müller cells are the major glial cell type of the retina, that provide homeostatic, metabolic and structural support to retinal neurons (Bringmann 2006).

Despite the absence of spontaneous regeneration in mammals, in vitro and in vivo studies demonstrated that mammalian Müller cells can also reprogram and produce new neurons in response to injury, provided treatment with appropriate growth factors or following overexpression of specific proneural genes (Salman 2021; García-García 2020; Lahne 2020; Wilken & Reh 2016).”

Regarding retinal regeneration of the embryonic retina in utero, we are not aware of any study. Of note, in the mouse, the retina is still in development at the end of embryogenesis, the majority of Müller cells (like photoreceptors and bipolar cells) being born at post-natal stages.

Since newborn photoreceptor cell are defective in crispants as well, did the authors try to rescue the defect by the coinjection of “inducible” rhodopsin or lipofection/electroporation of expression plasmids?

No, we have not tried to rescue the phenotype. In order to rescue post-embryonic photoreceptor production, we would need to target CMZ cells, which we have not succeeded yet by electroporation. This would however be very interesting in order to know whether rescued photoreceptors could survive in such degenerative context and restore some vision.

Discussion: I wonder about the fact that in frogs and in fish the pseudotetraploid possesses a better regenerative capacity than the diploid counterpart. It there a kind of dose-effect of the respective gene, different epigenetic regulation? Moreover, I wonder whether it is possible to perform single cell analysis of Müller cells for comparison.

This is a very interesting topic. scRNAseq on Müller cells could indeed contribute to address this question and we indeed plan to undertake some sequencing analysis. We believe however that this is beyond the scope of the present manuscript.

How much is known about endocytosis/autophagy/energy metabolism of the Müller glial cells (retinal cells) under regeneration? (AMPK, p62…, rab8)?

There are some recent interesting reports linking metabolism and Müller glial cells reprogramming and proliferative capacity, such as the manuscript from Andy Fischer’s lab on fatty acid metabolism (Cambell et al., Development 2022). Whether Müller cell metabolic differences between X. laevis and X. tropicalis could contribute to their different responses to retinal injury could thus be interesting to investigate in the future.

Typos:

Wording at the beginning of figure legend 3. …. (A) Typical retinal secTable 40. (?)

Wording line 5 of figure legend 5. …. Scheme 40. or 47. (?)

We apology for these two typos that were not present in our original manuscript and must have appeared during the manuscript conversion process.